# Weight–length relationships and Fulton's condition factors of skipjack tuna (*Katsuwonus pelamis*) in the western and central Pacific Ocean

Shaofei Jin[1,2,3], Xiaodong Yan[4], Heng Zhang[3] and Wei Fan[3]

[1] Key Laboratory of Regional Climate-Environment for Temperate East Asia (RCE-TEA), Chinese Academy of Sciences, Beijing, China
[2] University of Chinese Academy of Sciences, Beijing, China
[3] Key Laboratory of East China Sea & Oceanic Fishery Resources Exploitation and Utilization, Ministry of Agriculture, P. R. China (East China Sea fisheries Research Institute, Chinese Academy of Fishery Sciences), Shanghai, China
[4] State Key Laboratory of Earth Surface Processes and Resource Ecology, Beijing Normal University, Beijing, China

Corresponding authors
Shaofei Jin, jinsf@tea.ac.cn
Heng Zhang,
zhangziqian0601@163.com

## ABSTRACT

This paper describes the weight–length relationships (WLRs) and Fulton's condition factors (*K*) of skipjack tuna (*Katsuwonus pelamis*) in purse seine fisheries from three cruises in the western and central Pacific Ocean (WCPO): August–September 2009 (AS09), November–December 2012 (ND12), and June–July 2013 (JJ13). The fork length and weight of a total of 1678 specimens were measured. The results showed that the fork length of more than 70% of specimens was below 60 cm (76% in AS09, 87% in ND12, and 73% in JJ13). The coefficient *b* in the combined sex group was 3.367, 3.300 and 3.234 in JJ13, AS09 and ND12, respectively. The *b* values of WLRs when fork length was >60 cm were significantly less than 3 ($P = 0.062$), but when fork length was <60 cm they were significantly greater than 3 ($P = 0.028$). The *K* value ranges of JJ13, AS09 and ND12 in different fork length groups were 1.3–1.84 (1.62 ± 0.18), 1.57–2.02 (1.86 ± 0.15), and 1.44–1.78 (0.65 ± 0.13), respectively. Moreover, *K* values in different fork length classes for each cruise had one turning point: 60–65 cm for JJ13; 60–65 cm for ND12; and 55–60 cm for AS09. The results of this study provide basic information on the WLRs and *K* values of skipjack tuna in different seasons and growth phases in the WCPO, which are useful for fishery biologists and fishery managers.

## INTRODUCTION

Skipjack tuna (*Katsuwonus pelamis*) is a pelagic, high productivity species with a maximum age below 4.5 years (*Fromentin & Fonteneau, 2001*). As an important commercial species, it is mainly caught in tropical and subtropical waters of the Pacific, Atlantic, and Indian oceans. Catches of skipjack tuna comprise more than 70% of all tuna catches in the western and central Pacific Ocean (WCPO), where half of the world's tuna is caught

(*Food and Agricultural Organization of the United Nations, 2011*). In the WCPO, 86% of skipjack tuna are caught from the purse seine fishery (*Harley et al., 2011*). During fishery activities, some basic biological parameters (e.g., size, weight) are crucial for evaluating fishery sustainability and to assess stocks (*Fromentin & Fonteneau, 2001*; *Hampton, 2000*).

The weight–length relationship (WLR) and Fulton's condition factor ($K$) are two main parameters used in fishery research, and have been closely related since they were first proposed (*Froese, 2006*). The WRL is the relationship between weight and length for a given species, and can be used to estimate the growth pattern. Previous studies have considered the WLRs of skipjack tuna in the CPWO (*Wild & Hampton, 1993*; *Sun & Yeh, 2002*; *Sun et al., 2003*; *Froese & Pauly, 2014*); however, they mostly concentrated on the WLR of all skipjack tuna specimens caught. Thus, the information covered by different age/body classes was not clearly identified. Moreover, based on a recent WCPO skipjack tuna status report (*Harley et al., 2011*), small fish (40–60 cm; 1–2 years) dominated catches, except in 2005. However, little is currently known about the differences between biological parameters during different growth phases. Confusion may result because $a$ and $b$ (regressed parameters of the WLR) have been used to compare the differences among the different stages of one observation, and different *in situ* observations; specifically, because $a$ values can affect $b$ values, and higher $b$ values are associated with smaller $a$ values (*Froese, 2006*). The value of $K$ is calculated from the weight and length, and can be used to estimate changes in nutritional condition. Few studies in recent years have examined the $K$ value for skipjack tuna. Previous studies show that the $K$ values for other fish change seasonally (see *Froese, 2006* and references therein) and with growth phase. Skipjack tuna are known to be strongly affected by macro-marine conditions, e.g., El Niño and La Niña (*Lehodey et al., 1997*; *Lehodey et al., 2013*; *Loukos et al., 2003*); however, the same changes as in other fish species were not reported for skipjack tuna. Although both the WLR and $K$ can reflect the growth conditions of certain fish, no studies that combine analysis of the two indices have been reported. Thus, it remains unclear whether they possess consistent trends that reflect the population characteristics of skipjack tuna in the WCPO.

To address these research gaps, the present study provides a comprehensive analysis of these biological parameters (fork-length frequency, WLR, and $K$) in skipjack tuna sampled in different seasons and different growth phases in the WCPO.

## MATERIALS & METHODS

### Study area

Skipjack tuna were sampled during three cruises in the WCPO: August–September 2009 (AS09); November–December 2012 (ND12); and June–July 2013 (JJ13) (Fig. 1). All the sampling stations were followed by fishing stations, and the sampling vessels possessed the same stretched mesh sizes and the same purse seine nets as those governed by the WCPFC (Western and Central Pacific Fisheries Commission). Details of the vessels and their cruises are provided in Table 1. A total of 1678 specimens were measured.

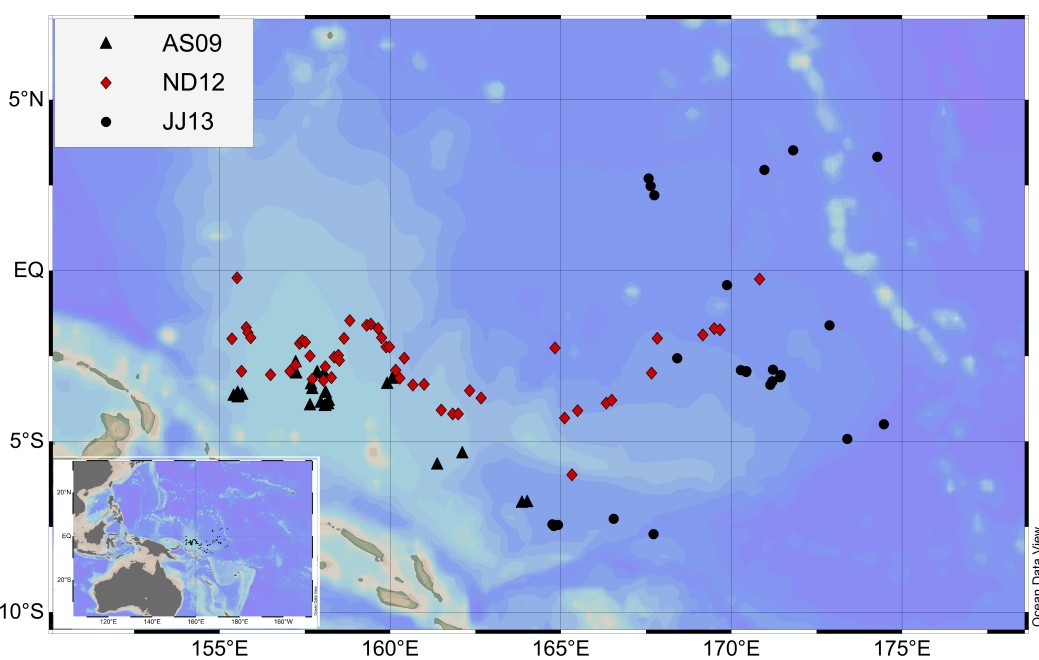

**Figure 1 Sampling map during the three cruises.** Black triangles, red diamonds and black circles indicate the stations in the August–September 2009 (AS09), November–December 2012 (ND12), and June–July 2013 (JJ13) cruises, respectively.

**Table 1 Details of the vessels and their cruises/data collection.**

| Cruise | Vessel name | Vessel length (m) | Vessel weight (tons) | No. stations | No. specimens |
| --- | --- | --- | --- | --- | --- |
| AS09 | JIN HUI NO.6 | 70 | 1,198 | 28 | 550 |
| ND12 | LOJET | 80 | 2,109 | 50 | 737 |
| JJ13 | LOMETO | 71 | 1,041 | 24 | 391 |

## Fork length and weight measurement

Fork length was measured from the tip of the snout to the end of the middle caudal fin rays; weight was measured as the total body weight ($\pm 5$ g).

## Fork length frequency

Fork length frequency was calculated with a 5 cm fork length interval between 30 cm and 75 cm. For each interval, the left boundary was closed. Taking the interval of 30–35 cm as an example, the fork length is from 30 cm (>30 cm) to 35 cm (35 cm). The frequency formula is as follows:

$$F_i = \frac{n_i}{N} \times 100\% \quad (i = 30\text{–}35\,\text{cm}, 35\text{–}40\,\text{cm}\ldots70\text{–}75\,\text{cm}) \tag{1}$$

where $F_i$ is the frequency for a certain interval, $n_i$ is the number of specimens in one fork length interval, and $N$ is the total number of specimens in one cruise.

## Weight–length relationship

The WLR was calculated using Eq. (2), where $a$ and $b$ are the coefficients, $L$ is the fork length (cm), and $W$ is the wet weight (g):

$$W = aL^b. \tag{2}$$

For the parameters in Eq. (2), the linear relationship between log $a$ (logarithmic value for $a$) and $b$ was used to determine whether the coefficients might be used in other studies. Coefficients that were far away from the regressed line produced by the coefficients combined from other studies were removed (*Froese, 2006*).

## Fulton's condition factor

The value of $K$ was calculated following *Froese (2006)*:

$$K = 100 * \frac{W}{L^3}. \tag{3}$$

For a given form, the volume can be calculated by multiplying one constant parameter by one measurable parameter cubic function, e.g. for a sphere, $V = 4/3\pi r^3$; for a cube, $V = l^3$. In general form, the volume can be written as $V = P \times M^3$, where $P$ is the constant parameter determined by the form and $M$ is a measurable length/diameter that has a high correlation with other measurable biometric parameters. For skipjack tuna, volume can be calculated as follows:

$$V = f(L)L^3 \tag{4}$$

where $f(L)$ is the constant parameter that has high correlation with the fork length. To connect the wet weight with the volume, one parameter representing density is needed. Some assumptions were made: (1) a mean density ($\rho$) for a certain fork length; (2) high relative linear correlations between fork length and maximum height ($H$) and between fork length and maximum width ($D$) (*Pornchaloempong, Sirisomboon & Nunak, 2012*; *Tičina et al., 2011*); and (3) bone shape did not change for a given fork length. Eq. (4) can now be rewritten as:

$$W = \rho * k * H * D * L = \rho * k * k_2 L * k_3 L * L \tag{5}$$

where $\rho$, $k$, $k_2$, and $k_3$ are the measurable parameters for a skipjack tuna of a given shape. Further, $H$ is a relatively stable parameter, $\rho$ is mean density, and $k$ is an ideal body shape parameter for a given bone shape. Thus, Eq. (5) can be rewritten as:

$$W = S * k_3 * L^3 = 100 * K * L^3 \tag{6}$$

where $S$ represents the consistent parameters for a given shape in a certain fork length interval. Based on the analysis processes above, higher $K$ values resulted in a higher $k_3$, indicating a thicker/fatter body.

**Table 2 Frequencies and proportions of different fork length groups in skipjack tuna.** 'Frequency' is the sample size, 'Proportion' is the result of Eq. (1), and bold numbers are the sum ('Frequency' and 'Proportion') of the fork-length group. AS09, ND12 and JJ13 represent the August–September cruise in 2009, the November–December cruise in 2012, and the June–July cruise in 2013, respectively.

| Fork length (cm) | AS09 | | ND12 | | JJ13 | |
|---|---|---|---|---|---|---|
| | Frequency | Proportion | Frequency | Proportion | Frequency | Proportion |
| CM | 550 | | 737 | | 391 | |
| <30 | 1 | 0.18% | 0 | 0.0% | 8 | 2.05% |
| 30–35 | 13 | 2.36% | 50 | 6.8% | 19 | 4.86% |
| 35–40 | 12 | 2.18% | 55 | 7.5% | 19 | 4.86% |
| **< 40** | **26** | **4.73%** | **105** | **14.2%** | **46** | **11.76%** |
| 40–45 | 103 | 18.73% | 138 | 18.7% | 45 | 11.51% |
| 45–50 | 99 | 18.00% | 264 | 35.8% | 65 | 16.62% |
| **40–50** | **202** | **36.73%** | **402** | **54.5%** | **110** | **28.13%** |
| 50–55 | 57 | 10.36% | 95 | 12.9% | 71 | 18.16% |
| 55–60 | 133 | 24.18% | 43 | 5.8% | 58 | 14.83% |
| **50–60** | **190** | **34.55%** | **138** | **18.7%** | **129** | **32.99%** |
| 60–65 | 125 | 22.73% | 56 | 7.6% | 42 | 10.74% |
| 65–70 | 7 | 1.27% | 25 | 3.4% | 49 | 12.53% |
| 70–75 | 0 | 0.00% | 11 | 1.5% | 15 | 3.84% |
| **>60** | **132** | **24.00%** | **92** | **12.5%** | **106** | **27.11%** |

## Statistical method

All statistical procedures, $t$-tests and one-way ANOVAs conducted in this paper were performed using R 3.1.2.

# RESULTS

## Frequency distribution of fork length

Table 2 shows the frequencies of skipjack tuna fork lengths from the three cruises. Fork lengths from 40 cm to 70 cm were dominant (about 84% of total specimens), and the proportion of fork lengths below 60 cm was 73% during the JJ13 cruise. The minimum fork length was 28 cm, and the maximum fork length was 74 cm (Table 2). During the AS09 cruise, 94% of fork lengths were between 40 cm and 65 cm, with 29 cm as the minimum fork length and 67 cm as the maximum fork length (Table 2). The proportion of fork lengths less than 60 cm was 76%. For the ND12 cruise, 67% of specimens were distributed between 40 cm and 55 cm, with a distribution peak (36%) in the 45–50 cm interval. The minimum and maximum fork lengths were 30 cm and 73 cm, respectively. Moreover, the proportion of fork lengths less than 60 cm was 87% (Table 2).

## Weight–length relationship

The WLRs of combined sex (CM) and different length intervals were calculated, with outlying thin or fat specimens excluded (Table 3). The WLR results compared among the

**Table 3 Weight–length relationships between fork length (cm) and wet weight (g) over the three cruises.** AS09, ND12 and JJ13 represent the August–September cruise in 2009, the November–December cruise in 2012, and the June–July cruise in 2013, respectively.

| Class | AS09 | | | ND12 | | | JJ13 | | |
|---|---|---|---|---|---|---|---|---|---|
| | $a$ | $b$ | $R^2$ | $a$ | $b$ | $R^2$ | $a$ | $b$ | $R^2$ |
| CM | 0.0058 | 3.2996 | 0.98 | 0.0066 | 3.2398 | 0.97 | 0.0039 | 3.3668 | 0.97 |
| <40 cm | 0.0084 | 3.2048 | 0.85 | 0.0049 | 3.3069 | 0.69 | 0.0072 | 3.1704 | 0.75 |
| 40–50 cm | 0.0026 | 3.5226 | 0.95 | 0.0031 | 3.4449 | 0.91 | 0.0184 | 2.9664 | 0.7 |
| 50–60 cm | 0.0064 | 3.2841 | 0.77 | 0.0199 | 2.9696 | 0.74 | 0.0426 | 2.7687 | 0.66 |
| >60 cm | 0.1681 | 2.481 | 0.59 | 0.9032 | 2.0441 | 0.61 | 0.1015 | 2.5835 | 0.68 |

**Notes.**

CM, combined sex; $a$, intercept; $b$, slope; $R^2$, coefficient of determination.

three cruises by CM was: $b$ (3.367 in JJ13) > $b$ (3.300 in AS09) > $b$ (3.234 in ND12). For different fork length classes, $b$ values showed no significant difference among all three cruises ($P = 0.745$, $F_{2,11} = 0.5$, one-way ANOVA). Additionally, all $b$ values with fork length >60 cm were significantly less than 3 ($P = 0.062$, $t$-test, $H_0$: $b = 3$; $H_1$: $b > 3$) with relatively weak correlation. However, the $b$ values of other classes were greater than 3 ($P = 0.028$, $t$-test, $H_0$: $b = 3$; $H_1$: $b > 3$) ($b$ values from all cruises). Furthermore, all correlations of the CM group were stronger than those of different fork length classes.

Parameters from the regressions tested for wiping off the outline data (*Froese, 2006*). Figure 2 shows the linear regression of the plot of log $a$ and $b$ has a high correlation ($R^2 = 0.996$). Compared to other similar studies, a high correlation was also found with data from this study and FishBase data (Fig. 2, solid line; sexed and unclear data excluded) (*Froese & Pauly, 2014*).

## Distributions of *K* values

Figure 3 illustrates the distributions of $K$ values from the three cruises. The $K$ value ranges of JJ13, AS09, and ND12 were 1.3–1.84 (1.62. ± 0.18), 1.57–2.02 (1.86. ± 0.15), and 1.44–1.78 (0.65 ± 0.13), respectively. The $K$ values in individual cruises showed an initial increasing trend over one fork length range, and then a declining trend over the following fork length range. Turning points were 60–65 cm, 60–65 cm, and 55–60 cm for JJ13, ND12 and AS09, respectively. The $K$ values of AS09 showed significant differences ($P = 0.01$, $F_{2,25} = 5.69$, one-way ANOVA, Tukey procedure of post-hoc analysis) with the values of the other cruises. All $K$ values of the specimens form the AS09 cruise were greater than those of the other two cruises. The $K$ values of ND12 were higher than those of JJ13 when the fork length was <60 cm, but the opposite trend was found when the fork length was >60 cm (Fig. 3).

## DISCUSSION

The parameters of WLRs indicate allometric growth of skipjack tuna (*Froese, 2006*), and these parameters are affected by many ecological and individual factors (*Perçin & Akyol, 2009*). While the WLR has been used for nearly 90 years (*Froese, 2006*), it has generally

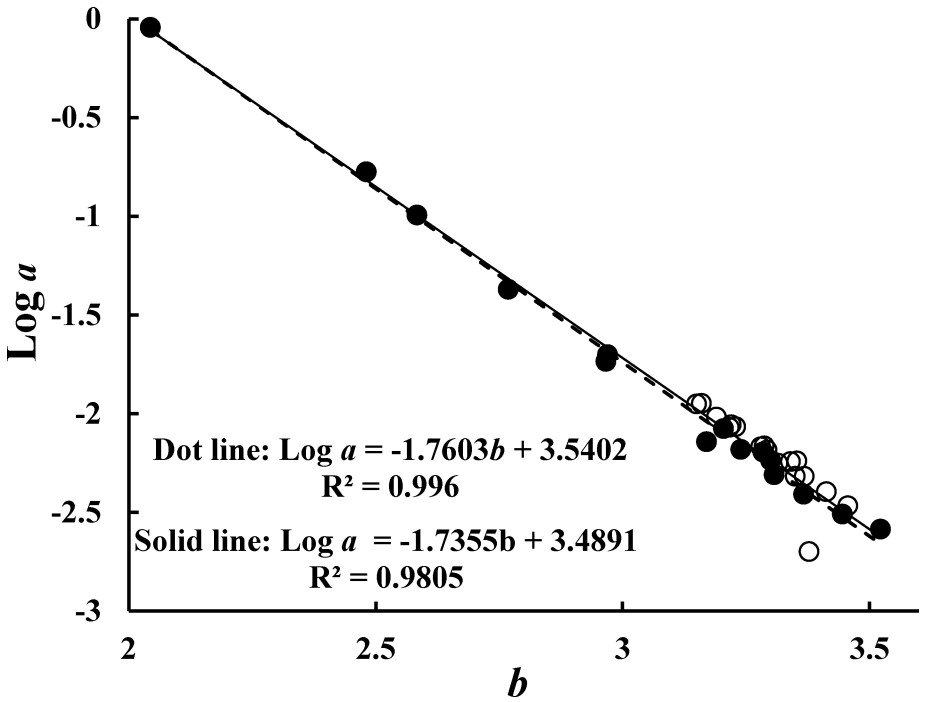

**Figure 2 Relationships between log *a* and *b*.** The dashed line is the linear regression line of data from this study (solid dots); the solid line is the linear regression line of combined data in this study and data excluding sexed and dubious data from FishBase (open dots).

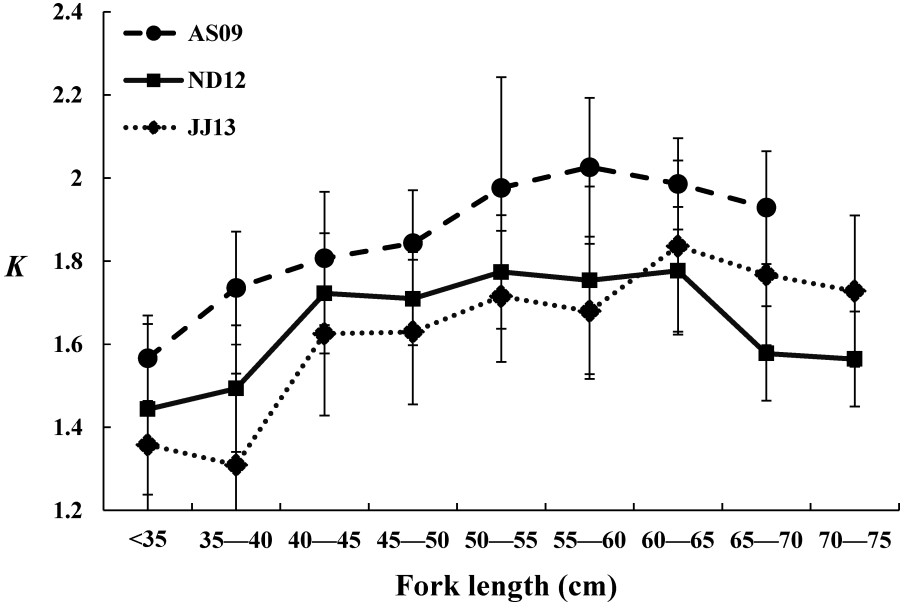

**Figure 3 Condition factor (*K*) per fork length (cm) class over all three cruises.** Error bars show the standard deviation. AS09, ND12, and JJ13 represent the August–September cruise in 2009, the November–December cruise in 2012, and the June–July cruise in 2013, respectively.

been the case that only the *b* values have been considered when comparisons are made. In this study, the WLRs of the CM class of skipjack tuna showed positive allometric growth (3.302 ± 0.064) for all specimens in the WCPO. *Wild & Hampton (1993)*, *Sun & Yeh (2002)*, and *Froese & Pauly (2014)* attained similar results. These results from the CM class indicate that the larger specimens were thicker than the smaller specimens (*Froese, 2006*). However, *b* values changed significantly (especially in the classes of fork length >60 cm) when different fork length classes were calculated, as recommended by *Froese (2006)*. The fact that *b* values were significantly less than 3 when fork length was >60 cm demonstrates an opposite understanding that larger fish are more elongated when fork length is >60 cm. Although our sample size was relative narrow compared with other reports based on over 1,000 samples (data from Fishbase, 2014), our study still obtained acceptable *a* and *b* values, as tested by plotting log *a* and *b* (*Froese, 2006*). Our results indicate it is more accurate to cover the growth pattern of skipjack tuna by obtaining the WLRs of its different growth phases.

K values were also used as a parameter to estimate the characteristics of fish body structures, such as *b* values for a certain fork length. However, arguments between *K* and *b* have lasted since 1920 (*Froese, 2006* and references therein). In this study, the *K* values of AS09 were larger than those of the other two cruises, indicating that the specimens caught by free-swimming schools in AS09 had thicker bodies than other fish in the same fork-length interval (Fig. 3). We acknowledge that empty stomachs may induce decreasing *K* values, as in bluefin tuna (*Thunnus thynnus*), reported by *Perçin & Akyol (2009)*. Furthermore, *Perçin & Akyol (2009)* suggested that health problems in large fish may reduce *K* values. Decreased *K* values of large fish were also found in our study, but the values were still greater than those of larval or premature fish (<40 cm) (Fig. 3). These results may indicate that the increasing sensitivity to ambient surroundings for larger/older and premature fish (*Stenseth et al., 2002*) may cause a decrease in their *K* values.

Over the three cruises, more than 70% of specimens were smaller than 60 cm (fork length); and *b* values when fork length was <60 cm were significantly greater than 3, while values when fork length was >60 cm were significantly less than 3. Similarly, *K* values had a turning point when the fork length was around 60 cm: when fork lengths were <60 cm, *K* values showed an increasing trend, while when fork lengths were >60 cm, *K* values decreased. This rule suggests that dividing the population structure for skipjack tuna into two stages (growth stage and old stage) could be a way to investigate the growth pattern for skipjack tuna that are younger than 2 years, and the sensitivity to environmental conditions for older fish. We suggest that *b* values should mainly be used for assessing growth rates due to their high rate of increase in the growth stage, while *K* values should be used for evaluating the sensitivities of fish to ambient factors or health conditions during the old stage.

## CONCLUSION

This paper provides information on the WLRs and *K* values in different fork-length classes in different seasons and growth stages for skipjack tuna in the WCPO. The work addresses the lack of studies on the basic parameters of skipjack tuna in different seasons and growth

phases over this region. The results will be not only useful for fishery research, but also for the fishery management commission in the WCPO. Although basic information on WLRs and $K$ values are provided, the reasons behind their changes require further study; the explicit relationship between morphological characteristics and environmental changes is yet to be revealed.

## ACKNOWLEDGEMENTS

The authors sincerely thank academic editor Professor María Ángeles Esteban and reviewers Professor Felipe Amezcua and Professor Martin Soto-Jimenez for their helpful comments on our manuscript. We also wish to express our deep appreciation to all the fishermen onboard *Jinhui NO. 6* vessel, *LOJET* vessel, and *LOMETO* vessel; we would not have been able to accomplish the field sampling work without their help.

### Funding

This study was funded by Natural Science Foundation of China (Grant No. 91425304), Shanghai Science and technology innovation action plan (Grant No. 12231203901), and National science and technology support program (Grant No. 2013BAD13B01). The funders had no role in study design, data collection and analysis, decision to publish, or preparation of the manuscript.

### Grant Disclosures

The following grant information was disclosed by the authors:
Natural Science Foundation of China: 91425304.
Shanghai Science and technology innovation action plan: 12231203901.
National science and technology support program: 2013BAD13B01.

### Competing Interests

The authors declare there are no competing interests.

### Author Contributions

- Shaofei Jin conceived and designed the experiments, performed the experiments, analyzed the data, contributed reagents/materials/analysis tools, wrote the paper, prepared figures and/or tables, reviewed drafts of the paper.
- Xiaodong Yan analyzed the data, reviewed drafts of the paper.
- Heng Zhang conceived and designed the experiments, performed the experiments, analyzed the data, contributed reagents/materials/analysis tools, reviewed drafts of the paper.
- Wei Fan conceived and designed the experiments, analyzed the data, reviewed drafts of the paper.

## Animal Ethics

The following information was supplied relating to ethical approvals (i.e., approving body and any reference numbers):

The study was carried out on commercial fishery vessels, governed and monitored by the WCPFC (Western and Central Pacific Fisheries Commission). All information related to skipjack tunas were reported to the WCPFC.

## Supplemental Information

Supplemental information for this article can be found online at http://dx.doi.org/10.7717/peerj.758#supplemental-information.

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
