# Peer review of "Weight–length relationships and Fulton’s condition factors of skipjack tuna (Katsuwonus pelamis) in the western and central Pacific Ocean"

_PeerJ, doi:10.7717/peerj.758_

## Round 0.1 · original submission · Major Revisions

· Academic Editor

Major Revisions

Dear Author,
The reviewers have commented on your paper. They indicated that it is not acceptable for publication in its present form. Furthermore, English should be improved and carefully revised.

·

Basic reporting

The English language is poorly written. Authors should ask for help of a colleague whose first language is English, in order to improve on this aspect.
Figures are not adequately labeled, for example, figures 1 and 3 do not explain the meaning of J-J, A-S and N-D. We can assume that, but the reader should be able to understand the figures, without reading the entire paper.

The introduction and background are not properly described. It is difficult to understand the goal and the aim of this work. If the authors are trying to do a fisheries related paper, then a better description of the fishery of this species should be given in this part, and what are the problems the authors are trying to solve with their work.

Experimental design

The authors have an interesting set of data, whoever they do not have a research questions. This work is rather descriptive, as there is not a working hypothesis. Because of this, the authors struggle to outline a proper discussion, as they are not really answering any question.
The methods are not adequately described; in the Results section the authors mention analyses that were never described in the Methods section. Also these are confuse, it is not clear to me why did they use the volume of the fish, and how this was estimated.

Validity of the findings

As there is not a proper research question, there cannot be proper conclusions. The amount of data is large, and some results are interesting, but the authors fail to explain the validity and relevance of these.

Additional comments

Authors need to establish a proper diagram of what they are trying to achieve with this work. They need to delineate a working hypothesis, and discuss and conclude around this.

·

Basic reporting

Sorry to be late to reply for the review of the manuscript “Combining Weight-Length Relationships and condition factors to estimate the population structure for Skipjack tuna in the Western and Central Pacific Ocean”.


The main goal was to investigate the fish population structure and growth progresses of Skipjack tuna by morphology parameters (length frequency,​ ​weight-length relationship and condition factor).

Although the science and the methods seem adequate, the manuscript is plagued of grammatical.

In spite the English is not my native language, I made a lot of corrections to improve the manuscript and I wrote several question that need to be cleared. I attached the revised manuscript.

Best

Experimental design

No comments

Validity of the findings

No comments

Additional comments

review of the manuscript “Combining Weight-Length Relationships and condition factors to estimate the population structure for Skipjack tuna in the Western and Central Pacific Ocean”.


The main goal was to investigate the fish population structure and growth progresses of Skipjack tuna by morphology parameters (length frequency,​ ​weight-length relationship and condition factor).

Although the science and the methods seem adequate, the manuscript is plagued of grammatical.

In spite the English is not my native language, I made a lot of corrections to improve the manuscript and I wrote several question that need to be cleared. I attached the revised manuscript.

Best

---

## Round 0.2 · accepted · Accept

· Academic Editor

Accept

The manuscript has been improved according the reviewers suggestions.